# The Discovery of Novel α_2a_ Adrenergic Receptor Agonists Only Coupling to Gαi/O Proteins by Virtual Screening

**DOI:** 10.3390/ijms25137233

**Published:** 2024-06-30

**Authors:** Peilan Zhou, Fengfeng Lu, Huili Zhu, Beibei Shi, Xiaoxuan Wang, Shiyang Sun, Yulei Li, Ruibin Su

**Affiliations:** State Key Laboratory of Toxicology and Medical Countermeasures, Beijing Key Laboratory of Neuropsychopharmacology, Beijing Institute of Pharmacology and Toxicology, 27th Taiping Road, Beijing 100850, China; lff8591465@163.com (F.L.); a123zhuzhujy@163.com (H.Z.); shigogogo123@163.com (B.S.); sepchutz0807@163.com (X.W.); noah97sun@163.com (S.S.); bettylpaine@163.com (Y.L.)

**Keywords:** α_2A_ adrenergic receptor, virtual screening, Gαi/o proteins

## Abstract

Most α_2_-AR agonists derived from dexmedetomidine have few structural differences between them and have no selectivity for α_2A/2B_-AR or Gi/Gs, which can lead to side effects in drugs. To obtain novel and potent α_2A_-AR agonists, we performed virtual screening for human α_2A_-AR and α_2B_-AR to find α_2A_-AR agonists with higher selectivity. Compound P300–2342 and its three analogs significantly decreased the locomotor activity of mice (*p* < 0.05). Furthermore, P300–2342 and its three analogs inhibited the binding of [^3^H] Rauwolscine with IC_50_ values of 7.72 ± 0.76 and 12.23 ± 0.11 μM, respectively, to α_2A_-AR and α_2B_-AR. In α_2A_-AR-HEK293 cells, P300–2342 decreased forskolin-stimulated cAMP production without increasing cAMP production, which indicated that P300–2342 activated α_2A_-AR with coupling to the Gαi/o pathway but without Gαs coupling. P300–2342 exhibited no agonist but slight antagonist activities in α_2B_-AR. Similar results were obtained for the analogs of P300–2342. The docking results showed that P300–2342 formed π-hydrogen bonds with Y394, V114 in α_2A_-AR, and V93 in α_2B_-AR. Three analogs of P300–2342 formed several π-hydrogen bonds with V114, Y196, F390 in α_2A_-AR, and V93 in α_2B_-AR. We believe that these molecules can serve as leads for the further optimization of α_2A_-AR agonists with potentially few side effects.

## 1. Introduction

Alpha2-adrenergic receptors (α_2_-AR) are integral membrane proteins that belong to the superfamily of G-protein-coupled receptors (GPCRs). These receptors regulate effector function via activation of multiple members of Go and Gi G-protein families [1]. Three subtypes (α2A/α2-C10, α2B/α2-C2, and α2C/α2-C4) are encoded by distinct genes and widely expressed in the brain. α_2A_-AR is most abundant in the rat locus coeruleus, but is also widely localized in the brain stem, cerebral cortex, septum, hypothalamus, hippocampus, and amygdala. α_2B_-AR was observed only in the thalamus, while α_2C_-AR was mainly distributed in the basal ganglia, olfactory tubercle, hippocampus, and cerebral cortex [2]. α_2A_-AR is the predominant α_2_-AR subtype in the central nervous system, which prejunctionally inhibits the release of sympathetic, endogenous neurotransmitters such as norepinephrine and epinephrine. They mediate the antihypertensive and the sedative effects of α_2_-AR agonists [3,4]. The postjunctional α_2_-AR found in the peripheral tissues has been demonstrated to induce vasoconstriction and increase arterial pressure in various mammalian species [5,6]. The α_2B_-AR subtype is mainly distributed in the periphery, including lung, kidney, liver, heart, and vascular tissues. It appears to have a dominant role in eliciting the immediate or salt-loading hypertensive response to α_2_-AR agonists [7,8,9]. α_2C_-AR regulates the release of norepinephrine at low norepinephrine concentrations, in contrast to α_2A_-AR regulation at high norepinephrine concentrations [3]. The amount of α_2C_-AR is lower than that of α_2A_-AR in the brain. The autoinhibition mediated by α_2C_-AR in α_2C_-KO mice could be substituted by α_2A_-AR for the elusive character of α_2C_-AR [10].

A large amount of structural activity data has emerged in pursuit of the specific agonists and antagonists of α_2_-AR subtypes for potential therapeutics since the discovery of the three α_2_-AR subtypes. Exceptionally, α_2A_-AR has a dual pharmacological effect in that it simultaneously couples to Gi and Gs to inhibit or stimulate adenylyl cyclase (AC) activity [11]. At low agonist concentrations, α_2A_-AR mainly couples to Gi, whereas at high concentrations, Gs coupling dominates. This unusual dual effect has not been well understood for any GPCR. The Gi/o inhibitor pertussis toxin (PTX) pre-treatment inhibited DMED-induced sedation [12], whereas the Gs inhibitor had no effect on the sedation induced by DMED (data not published). The Gs coupling action of these partial agonists of α_2A_-AR may not be related to sedation, and the effects remain incompletely understood. α_2A_-AR agonists are mainly imidazole ring compounds that can interact with D113 by a salt bridge, and they have no selectivity for α_2A/2B_-AR or Gi/Gs [13]. To enhance the therapeutic potency of α_2_ agonists as antihypertensive and sedative drugs, compounds should bind to α_2B_-AR with lower affinity or have no coupling to Gs proteins at high concentrations.

Computer-aided drug design has been widely used to discover novel and potent lead compounds. This study combined crystal structures for human α_2A_-AR [14] and α_2B_-AR [15] with virtual screening to obtain highly selective α_2A_-AR agonists as potential sedatives. The ChemDiv compound library (1,535,478 compounds) was screened, and 1023 compounds were found that could bind to α_2A_-AR with high affinity but low affinity to α_2B_-AR. We selected 105 compounds with good druggability according to the virtual screening; these molecules were studied in the loss of righting reflex (LORR) of C57/B6 mice. Compound P300–2342 and three analogs had a sedative effect after α_2A_-AR activation, as did the Gi/o-coupled pathway with the inhibition of AC-cyclic adenosine monophosphate (cAMP) accumulation. These compounds could serve as leads for sedation by α_2A_-AR activation with potentially few side effects.

## 2. Results

### 2.1. Analysis of the Molecular Docking Results

All 50,679,311 molecular conformations generated by the Chemdiv compounds library (Version 2019) were docked to α_2A_-AR, and the top 100,000 compounds with high affinity were then docked to α_2B_-AR. The molecular weight distribution of these compounds ranged from 216 Dal to 767 Dal. The binding energy [ChemGauss Score (CGS)] of these compounds to α_2A_-AR ranged from −22 to −16 kcal/mol, and ranged from −19 to −8 kcal/mol for α_2B_-AR. A correlation analysis showed that the binding affinity of these compounds for the α_2A_-AR receptor did not correlate to that for α_2B_-AR [Appendix A], which made it possible to select the compounds that may bind to α_2A_-AR with high affinity but not to α_2B_-AR or with low binding affinity. The binding energies of the selected compounds to α_2A_-AR or α_2B_-AR ranged from −22 to −16 kcal/mol or −14 to −8 kcal/mol, respectively. In addition, compound C593–0297 was exceptionally selected with the binding energy −14.85 kcal/mol to α_2B_-AR [red dot in Appendix A]. Finally, 1023 compounds from the Chemdiv database were screened for druggability filtrates.

### 2.2. Properties of the Selected Compounds

StarDrop software (6.5.0) was used to analyze the properties of 1023 compounds including water solubility (logS), the lipid–water distribution coefficient (logP), molecular weight, molecular flexibility, the hydrogen bond, the topological polar surface access area, the cytochrome P450 2C9 (CYP2C9) enzyme degradation level, the human *ether-à-go-go-related gene* (hERG) value, oral availability, drug interaction risk (2D6), and other indicators. The summarized score (SS) was calculated based on these properties. The SS values of these compounds ranged from the largest, 0.4379, to the smallest, 0.0023. According to the SS values, the top 500 compounds with the highest SS scores were selected for the following analysis.

A cluster analysis was performed by a maximum common substructure algorithm to obtain structural diversity of candidate compounds; the similarity threshold was set to 0.5, which divided the 500 compounds into 67 clusters. The correlation plots between the SS scores and other properties such as binding affinity, logS, and logP were analyzed [Appendix A]. By analyzing the structural diversity, binding affinity, and druggability scores, 140 compounds with the best SS values and diverse structures were kept [red in Appendix A], which were further classified into 21 clusters with an enhanced similarity threshold (0.7) [Figure 1]. Most of the categories contained only two–three compounds, but cluster 2, 4, 11, and 12 contained more compounds. The compounds from cluster 4 and 16 showed high binding affinity to α_2A_-AR [Figure 1A]. Compared with other clusters, compounds 4 and 10 also bound to α_2B_-AR [Figure 1B]. Compounds from cluster 7, 9 and 14 may have had lower water solubility [Figure 1C].

### 2.3. The Effects of Compounds on the Lorr and Locomotor Activity in Mice

We studied 105 compounds with good druggability in the virtual screening chemicals in the LORR of C57/B6 mice. DMED (432 nmol/kg, i.p.) induced 50% LORR in mice. After intraventricular administration, 28 compounds increased and 4 compounds decreased the percent of LORR induced by DMED (432 nmol/kg, i.p.). A total of 73 compounds displayed no effect on the LORR induced by DMED (432 nmol/kg, i.p.) [Figure 2A]. The 28 compounds (100 nmol/mice, i.c.v.) were further studied for locomotor activity. Among these 28 compounds, 4 compounds significantly inhibited the locomotor activity of mice (*p* < 0.05) [Figure 2B]. Only the inhibition of P300–2342 (10 nmol/mice, i.c.v.) in the mice locomotor activity could be antagonized by the α_2_-AR antagonist atipamezole (0.2 mg/kg, i.p.) [Figure 2C]. A total of 13 analogs of P300–2342 were also studied through the locomotor activity of mice. There were five analogs, P300–2344, P300–1500, P300–2297, T226–1918, and T226–1970 (10 nmol/mice, i.c.v.), that inhibited the mice locomotor activity 30%–60% compared to that of the solvent [Figure 2D].

### 2.4. The Interaction of P300–2342 or Analogs with A_2a_-Ar or A_2b_-Ar

P300–2342 had affinity for α_2A_-AR and α_2B_-AR, inhibiting the binding of [^3^H] Rauwolscine to the adrenergic receptors with IC_50_ 7.72 ± 0.76 and 12.23 ± 0.11 μM, respectively. The maximum binding inhibition of [^3^H] Rauwolscine was 83% for α_2A_-AR and 71% for α_2B_-AR [Figure 3A,B]. The P300–2342 analogs T226–1970, P300–2344, and P300–1500 concentration-dependently inhibited the binding of [^3^H] Rauwolscine to α_2A_-AR with maximum inhibition at 85%, 91%, and 90%, respectively. The half maximal inhibitory concentrations (IC_50_) of T226–1970, P300–2344, and P300–1500 inhibiting the binding of [^3^H] Rauwolscine to the α_2A_-adrenergic receptors were 19.54, 11.34, and 9.66 μM, respectively [Figure 3C]. T226–1970, P300–2344, and P300–1500 also concentration-dependently inhibited the binding of [^3^H] Rauwolscine to α_2B_-AR with a maximum inhibition of 94%, 86%, and 85%, respectively, with IC_50_s of 8.56, 15.91, and 19.05 μM, respectively [Figure 3D]. P300–2297 and T226–1918 had no inhibition on the binding of [^3^H] Rauwolscine to α_2A_- or α_2B_-AR. The α_2_ agonists DMED and UK 14304 inhibited the binding of [^3^H] Rauwolscine to α_2A_-adrenergic receptors with IC_50_s of 80.18 and 293.56 nM, respectively. DMED and UK14304 inhibited the binding of [^3^H] Rauwolscine to α_2B_-AR with IC_50_ of 14.63 and 550.78 nM, respectively [Figure 3B].

To investigate the binding mode of identified compounds to α_2A_-AR, the interaction of compounds with α_2A_-AR were studied using OpenEye software (3.2.0.2). P300–2342 formed a π-hydrogen bond with Y394 in the sixth transmembrane helix and V114 in the third transmembrane helix of α_2A_-AR with a rigid docking score of –17.47 kcal/mol [Figure 4(A1,A2)]. P300–2342 also formed a π-hydrogen bond with V93 in α_2B_-AR with a rigid docking score of −11.91 kcal/mol [Figure 4(A3)], which was lower than that of α_2A_-AR [Figure 4(A2)]. T226–1970 formed one π-hydrogen bond with F390 in the sixth transmembrane helix and two π-hydrogen bonds with L110, V114 in the third transmembrane helix of α_2A_-AR. The T226–1970 side chain amino formed a hydrogen bond with D113 in the third transmembrane helix of α_2A_-AR [Figure 4(B1,B2)]. P300–1500 formed π-hydrogen bonds with V114, F390 in the third and sixth transmembrane helix, and formed π-hydrogen bonds with Y196 in extracellular loop2 of α_2A_-AR. The P300–1500 side chain formed a hydrogen bond with D113 in the third transmembrane helix of α_2A_-AR [Figure 4(C1,C2)]. P300–2344 formed a π-hydrogen bond with V114 and a hydrogen bond with D113 of α_2A_-AR [Figure 4(D1,D2)]. T226–1970, P300–1500, and P300–2344 all formed π-hydrogen bonds with V93 in the C-terminal of the second transmembrane helix of α_2B_-AR [Figure 4(B3,C3,D3)].

### 2.5. Cyclic Adenosine Monophosphate Assay

In HEK293 cells co-transfected with the pGloSensor-22F cAMP plasmid and pCMV6 Entry-Flag-α_2A_-AR, DMED (10^−13^–10^−9^M) concentration-dependently decreased cAMP accumulation stimulated by forskolin (FSK) (10^−5^M). In contrast, DMED (10^−8^–10^−5^M) concentration-dependently increased cAMP accumulation. P300–2342 (10^−10^–10^−5^M) concentration-dependently decreased cAMP accumulation without increasing cAMP accumulation. The effects of P300–2342 and DMED on the cAMP accumulation were antagonized by the α_2A_-AR selective antagonist BRL44408 (1 μM). The decrease in cAMP accumulation induced by P300–2342 and DMED was abolished by a 500 ng/mL pretreatment of pertussis toxin (PTX) for 24 h [Figure 5A]. DMED (10^−10^–10^−7^M) concentration-dependently increased cAMP accumulation in HEK293 cells transfected with α_2B_-AR. In contrast, P300–2342 (10^−10^–10^−5^M) had no effect on cAMP accumulation [Figure 5B]. P300–2342 formed π-hydrogen bonds with V114 in the third and Y394 in the sixth transmembrane helixes of α_2A_-AR. In the α_2A_-AR mutant V114A (α_2A_V114A-AR), or Y394A (α_2A_Y394A-AR)-expressed cells, P300–2342 did not decrease cAMP production. Similarly, DMED did not decrease cAMP production in the α_2A_V114A-AR or α_2A_Y394A-AR transfected cells. However, DMED increased cAMP production in α_2A_V114A-AR or α_2A_Y394A-AR compared to that of α_2A_-AR wild type [Figure 5C,D]. These results indicated that V114 and Y394 in α_2A_-AR play important roles in the agonist activation of α_2A_-AR that couples to the G_αi_ protein.

As well as P300–2342, the analogs T226–1970, P300–2344, and P300–1500 decreased cAMP production without increasing cAMP accumulation in α_2A_-AR-HEK293 cells [Figure 6A]. T226–1970, P300–2344, and P300–1500 also had no activation in α_2B_-AR [Figure 6B]. In α_2B_-AR-HEK293 cells, the α_2B_-AR selective antagonist ARC239 (10 μM) decreased the cAMP accumulation induced by DMED and made the dose–response curve shift to the right. P300–2342 and T226–1970 (10 μM) slightly decreased the cAMP accumulation induced by DMED, whereas P300–2344 and P300–1500 (10 μM) had no effects on the DMED-stimulated cAMP [Figure 6C].

## 3. Discussion

The selective α_2_-AR subtype antagonist and α_2_-AR subtype knockout mice demonstrated α_2A_-AR-subtype-mediated sedation induced by α_2_-AR agonists [16]. Presynaptic α_2A_-AR activation of Gi/o can inhibit the activity of adenylate cyclase with the attenuation of cAMP production, activation of K^+^ conduction, acceleration of Na^+^/H^+^ exchange, and inhibition of Ca^2+^ channels [17]. As the classic GPCR, α_2A_-AR has seven transmembrane segments, three extracellular loops, and three intracellular loops. Each transmembrane segment consists of 20 to 26 amino acids forming an α helix, in which the hydrophobic amino acids form a “pocket”, which determines the specificity of binding to the ligand. The second and third intracellular loops are related to the G protein coupling of the receptor. The N-terminal part of the third intracellular loop and some amino acid residues of the second intracellular loop determine the specificity of the interaction between the receptor and the G protein. The C-terminal part of the third intracellular loop is also involved in the interaction between the receptor and the G protein, but it is mainly related to the coupling efficiency of the receptor and the G protein [11,18].

The Chemdiv database (Version 2019), containing 1,535,478 small molecule compounds with diverse backbones, was used for virtual screening. To ensure the global conformation of small molecules in the virtual screening process, the plug-in omega in OpenEye software was used to generate small molecule conformations. On average, each small molecule generated about 50 conformations. We converted the Chemdiv molecules into 50,679,311 molecular conformations and docked the compounds with α_2A_-AR crystal structures, and the top 100,000 were then docked to α_2B_-AR crystal structures. There was no obvious correlation in the compound affinities to α_2A_-AR and α_2B_-AR [Appendix A]. The Chemdiv database binding the α_2A_-AR receptor with high affinity but with low binding affinity to α_2B_-AR allowed us to select 1023 compounds for property analysis. The 500 compounds with the highest SS scores (logS, logP, molecular weight, molecular flexibility, hydrogen bond, topological polar surface area (TPSA), CYP2C9 degradation, hERG inhibition, HIA, 2D6, etc.) were selected for structural diversity analysis. The 140 compounds with the best SS values and diverse structures were further classified into 21 categories with an enhanced similarity threshold (0.7) [Figure 1].

In our study, approximately three quarters of the 105 compounds with the best SS values had no sedation or anti-sedative effects, one quarter of the compounds had sedative effects, and a few compounds had anti-sedative effects [Figure 2A, Appendix A]. Most of the small molecular compounds interacted with D113 in the third transmembrane helix and E189 in the second extracellular loop. At the same time, N93 and V114 were also involved in the binding of most compounds. In addition, 17 amino acid sites, including W99, C106, Y109, C117, T118, S165, C188, I190, D192, Y196, S200, F390, F391, Y394, K409, F412, and Y416, were involved in the interaction of the compounds and α_2A_-AR in the present study. D113 in the third transmembrane domain (TM3) is the conservative ligand-contacting residue in the aminergic receptor including α_1_, α_2_, and the β adrenergic receptor [19]. D113N inhibits the binding of [^3^H] yohimbine to α_2_-adrenergic receptors with a small decrease in cAMP accumulation stimulated by forskolin and the potentiation of forskolin-stimulated cAMP accumulation [13]. S200 in the fifth transmembrane domain (TM5) of hα_2_-AR involved in the ligand binding pocket plays an important role in epinephrine and UK-14304 binding and response [13,20]. Qu et al. found twelve residues (D113, V114, S200, C201, S204, W387, F390, F391, Y394, F408, F412, and Y416) involved in the binding pocket of α_2A_-AR by a co-crystallized partial agonist and antagonist. These four residues, D113, F390, F412, and Y416, are involved in the formation of an aromatic cage in both α_2A_-AR and α_2B_-AR [14]. In our study, Y394 in α_2A_-AR formed a π-hydrogen bond with an imidazoline ring in compound P300–2342. Y394 formed hydrogen bonds with the α_2A_-AR agonists epinephrine, norepinephrine, and UK14,304 [14]. V114 in the third transmembrane helix of α_2A_-AR also formed a π-hydrogen bond with a phenyl ring in P300–2342. P300–2342 also formed a π-hydrogen bond with Val93 in α_2B_-AR receptors (Figure 4(A3)).

Similar to previous reports, α_2A_-AR wild type (WT) simultaneously coupled to both the Gi and Gs proteins [21] when stimulated by the agonists UK14,304 and DMED (Figure 5A). However, α_2A_-AR WT activated by P300–2342 only coupled to Gi proteins (Figure 5A and Figure 6A). In contrast to α_2A_-AR and Y394N in previous reports [14], the mutant α_2A_-AR Y394A lost the coupling to Gi proteins and only coupled to Gs proteins with DMED treatment [Figure 5D]. In the epinephrine-bound β2-AR, epinephrine formed hydrogen bonds with S204 and N293 (6.55), which play an important role in the Gs activation of β2-AR. N293 is relative to Y394 (6.55) in α_2A_-AR [22]. The polar amino acids Y394 or N394 in α_2A_-AR are important to Gi protein signals under agonist addition. In contrast, the nonpolar amino acid A394 mutant α_2A_-AR only activated Gs protein signals in our study.

GPCR signaling is well-demonstrated according to the progress in GPCR structural biology. The GPCR cytoplasmic end of the seven transmembrane helix domain is closed in the inactive states [23,24]. The sixth transmembrane helix plays an important role in GPCR-Gi activation [25,26,27]. In the present study, α_2A_-AR Y394A in the mutant sixth transmembrane helix lost Gi protein activation upon agonist addition, in accordance with the previous view. Except for the amino in transmembrane helix 6, the three serines (S200, S201, and S204) in transmembrane helix 5 and D130 in transmembrane helix 3 are involved in the agonist interaction with β2-AR [22,28,29]. These amino acids are also involved in α_2_-AR interactions with catecholamines [30], which might play an important role in Gs protein coupling. P300–2342 formed a π-hydrogen bond with Y394 in the sixth transmembrane helix and V114 in the third transmembrane helix in the binding pockets of α_2A_-AR. The interaction between P300–2342 and α_2A_-AR could not make the receptor couple to the Gs protein. Consistent with the α_2A_-ARY394A mutant, α_2A_-AR V114A only coupled to Gs proteins after DMED addition [Figure 6C]. P300–2342 lost its coupling to Gi proteins in α_2A_-AR, Y394A, α_2A_-AR, and the V114A mutant, which indicated that V114 in the third and Y394 in the sixth transmembrane helixes play important roles in Gi protein coupling (Figure 6C,D). Although P300–2342 formed a π-hydrogen bond with V93 in α_2B_-AR and inhibited the binding of [^3^H] Rauwolscine to α_2B_-AR, P300–2342 had no activation in α_2B_-AR coupling to Gs proteins. V93 is a conserved site in α_1_, α_2_, and β-AR [15]. Yuan et al. found V93 in the third transmembrane helix involved in the binding pocket of DMED in α_2B_-AR, but V93 had no interaction with DMED [15]. V93 might not to be the key site for the agonist activation. The P300–2342 interacting sites Y394, V114 in α_2A_-AR, and V93 in α_2B_-AR contributed to the uncoupling to Gs proteins, which prevented the unwanted Gs coupling activity. The P300–2342 analogs T226–1970, P300–2344, and P300–1500 also formed π-hydrogen bonds or hydrogen bonds with α_2A_-AR and α_2B_-AR, and then inhibited the binding of [^3^H] Rauwolscine to α_2A_-AR and α_2B_-AR. α_2A_-AR only coupled to Gi proteins following the activation of three analogs of P300–2342. T226–1970, P300–2344, and P300–1500 also had no activation to α_2B_-AR (Figure 6B). The structure of P300–2342 and the three analogs is more complex than that of the α_2A_-AR agonists like epinephrine, dexmedetomidine, and clonidine [Figure 2]. These compounds were also larger than the α_2A_-AR full agonist UK14304. P300–2342 and its analogs had several benzene moieties with a long polar residue. These large molecular compounds were hydrophobic and obstructed the interaction with the residues in α_2A_-AR. Therefore, α_2A_-AR could not couple to Gs proteins after being activated by P300–2342 and its analogs. P300–2342 and its analogs could serve as lead compounds with sedative effects without the vascular side effects. We speculate that the future development of compounds capable of selectively activating α_2A_-AR while blocking α_2B_-AR may further improve our capability to treat hypertension, ischemic heart disease, and heart failure.

## 4. Conclusions

Most α_2_-AR agonists are imidazole ring compounds that have no selectivity for α_2A/2B_-AR or Gi/Gs, potentially resulting in drug side effects. In this study, we performed a structure-based virtual screening strategy to identify the selective α_2A_-AR agonists. We used virtual screening and druggability filtering for the ChemDiv library. We found that 1023 compounds may bind to α_2A_-AR with high affinity but with low affinity to α_2B_-AR. Among them, 105 compounds with good druggability according to virtual screening were purchased and studied in the LORR of C57/B6 mice. Ligand binding assays showed that compound P300–2342 selectively activated α_2A_-AR, and the cAMP assay indicated that P300–2342 could only couple to the Gi/o protein signaling pathway for sedation. Compound P300–2342 may serve as a lead compound for sedatives by activation of α_2A_-AR with potentially few side effects. Our research can provide a basis for developing novel selective agonists.

## 5. Materials and Methods

### 5.1. Protein Preparation for Humanα_2_-AR Proteins and Binding Pocket Identification

The 3D structure of α_2A_-AR in complex with a partial agonist E39 (pdbid:6kuy) is shown in Appendix A, where the ligand binding area is in the extracellular domain. The ligand binding area contained four adjacent hydrophobic pockets [Appendix A]. Similarly, the 3D structure of α_2B_-AR in complex with agonist DMED (pdbid:6k42) is shown in Appendix A, with seven transmembrane regions and an extracellular domain, which was highly similar to the structure of α_2A_-AR subtype (all atoms RMSD = 4.27 Å). The ligand-binding site was located in the extracellular domain, which is consistent with the ligand-binding region of the α_2A_-AR subtype. Two adjacent molecular binding pockets were selected at the ligand site in crystal structures by the MOE package [Appendix A]. In order to identify the compounds which could specifically bind to the α_2A_-AR subtype but not α_2B_-AR, the relevant pockets from both subtypes were chosen to perform the structure-based virtual screening.

### 5.2. Virtual Screening

The crystal structure of α_2A_-AR and α_2B_-AR were protonated and optimized by MOE QuickPrep (2019.0102). The optimized structures were then converted to the formatted receptor files that defined all the above-mentioned residues as the docking area by using the appropriate (apopdb2receptor) module of OpenEye software (Release 3.2.0.2). The docking areas of α_2A_-AR and α_2B_-AR were a rectangle with a box size of 27.67 Å × 19.33Å × 22.33 Å and 28.00 Å × 23.67 Å × 23.67 Å, respectively.

The Chemdiv compound library was selected for virtual screening, which contained 1,535,478 compounds. FRED performs rigid docking, so OMEGA2 was used with all default parameters to search for as many conformations as possible for each compound. OEDocking (Release 3.0) was used to dock all conformations of compounds to α_2A_-AR, with the parameters -save_component_scores: true; -hitlist_size: 100,000; -docked_molecule_file: sdf. The top 100,000 compounds for α_2A_-AR were docked to the binding pocket of α_2B_-AR, and multiple conformations were also generated by OMEGA. StarDrop (Release 6.6.4) was used to calculate the structural diversity based on a maximum common substructure algorithm to cluster compounds containing a significant common substructure. The molecular properties such as logS, HIA category, logP, hERG, 2D6, 2C9 affinity, P-gp category, PPB90, and BBB permeability were used as a scoring function to evaluate the druggability of each molecule.

### 5.3. Loss of Righting Reflex (LORR) and Locomotor Activity

#### 5.3.1. Materials

C57BL/6 mice (20–22 g) were purchased from the SPF biotechnology co., LTD (Beijing, China). All the animals were housed in an alternating 12 h light/12 h dark cycle room, which was maintained in a climate-controlled environment (25 ± 1 °C). Animals had free access to water and food. All the experimental procedures were reviewed and approved by the ethics committee and institutional animal care and use committee of the Beijing Institute of Pharmacology and Toxicology, Beijing, China (IACUC of AMMS–06–2017–001). Animals were assigned to groups randomly before testing. Testing occurred within the same approximate time of day between experiments. The experimenter was blinded to treatment throughout the course of behavioral testing.

Drugs were purchased from various vendors. Dexmedetomidine was purchased from (Dexinjia Biotechnology, Jinan, China). BRL44408 was purchased from (Sigma, Louis, MO, USA). Forskolin was purchased from (Sigma, Louis, MO, USA). UK 14,304 was purchased from (Tocris, Bristol, UK). [^3^H] Rauwolscine was purchased from (PerkinElmer, Waltham, MA, USA). The selected compounds were purchased from (Topscience, Shanghai, China). Dexmedetomidine and BRL44408 were dissolved in sterile physiological saline (0.9% NaCl) to the final concentration used in this study. The selected compounds were dissolved in 5% DMSO.

#### 5.3.2. LORR and Locomotor Activity

The LORR in C57/B6 mice were used to study the effects of screening compounds. Mice were pre-treated with compounds (100 nmol/mouse, i.c.v.) 15 min prior to DMED administration (432 nmol/kg, i.p.). The righting reflex was deemed to be “lost” if the mouse failed to right itself within 60 s of being placed on its back, as described in our previous study [31]. The effects of the compounds on the spontaneous locomotor activity of mice were detected using a LabState video tracking system (Anilab Scientific Instruments Co., Ltd., Ningbo, China). The mice were placed in acrylic boxes (40 cm × 40 cm × 35 cm) for 30–60 min before compound treatment on day 1. On day 2, compounds (10 or 100 nmol) were injected into the cerebral ventricles (i.c.v.) of the mice 30 min before being placed in acrylic boxes. The locomotor activity of each mouse was defined by the horizontal distance it travelled (in centimeters) over 30–60 min [32].

### 5.4. Ligand-Binding Assay

The competitive inhibition of [^3^H] Rauwolscine (2.5 nmol/L) binding to human α_2A_-AR or α_2B_-AR was performed in the presence of various concentrations of P300–2342, T226–1970, T226–1918, P300–2344, P300–2297, P300–1500, UK 14304, and DMED. CHO-K1 cells expressing human α_2A_-AR or α_2B_-AR membranes (2 μg, ES-030-M400UA/NET722250UC, PerkinElmer) were incubated at 27 °C for 1 h in 50 mM Tris-HCl, 1 mM EDTA buffer with 2.5 nM [^3^H] Rauwolscine in the presence or absence of ten concentrations of competing ligands ranging from 1.3 × 10^−9^ to 1.0 × 10^−4^. The reactions were transferred to the pretreated UNIFILTER-96 GF/B plates (PerkinElmer) and rinsed four times with 50 mM Tris-HCl by Universal Harvest (PerkinElmer). The plates were dried at 55 °C for 10 min, and 40 μL ULTIMA GOLD (PerkinElmer) scintillation fluid was added to each well. Radioactivity retained in each sample was counted in a Microbeta2 (PerkinElmer). The binding inhibition % = (average value of negative control-CPM)/(average value of negative control-average value of positive control) × 100.

### 5.5. cAMP Assay

According to the GloSensor cAMP biosensor (Promega, Shanghai, China) manufacturer protocols, HEK293 cells were co-transfected with the pGloSensor-22F cAMP plasmid (Promega, E1171) and pCMV6 Entry-Flag-α_2A_-AR. After 24 h, the cells were seeded in white 96-well plates. The next day, the cell media were replaced by 90 μL of fresh DMEM with 2% v/v GloSensor cAMP reagent (Promega, E1290) and incubated for 60 min at 37 °C. Before drug stimulation, an initial measurement of the baseline signal was detected. Then cells were stimulated with 10^−12^, M-10^−5^, M, P300–2342, T226–1970, P300–2344, P300–1500, or DMED for 10 min and then were stimulated with forskolin (10 μM) for 15 min with or without α_2A_-AR antagonist BRL44408 (10 μM) pretreatment. PTX 500 ng/mL selectively ablated receptor Gi coupling [32]. PTX 500 ng/mL pretreatment for 24 h was used to assay the chemical effects on Gi coupling following α_2A_-AR activation. Two mutations, V114A and Y394A were introduced into the α_2A_-AR genes, respectively, using standard QuickChange PCR. The cAMP accumulations induced by DMED or P300–2342 were also studied in HEK293 cells co-transfected with the pGloSensor-22F cAMP plasmid and pCMV6 Entry-Flag-α_2A_V114A-AR or α_2A_Y394A-AR with forskolin (10 μM) stimulation. HEK293 cells co-transfected with the pGloSensor-22F cAMP plasmid and pCMV6 Entry-Flag-α_2B_-AR were stimulated with P300–2342, T226–1970, P300–2344, P300–1500, or DMED to detect the cAMP accumulation. In each experiment, the cAMP assay used pcDNA3.1 myc/hisB transfected cells as a negative control and cells stimulated with 10 μM of forskolin as a positive control. The signal was read using a Victor 2D Instrument (PerkinElmer) at 675 nm. The production of cAMP% was calculated (the signal value after chemicals addition–baseline signal value)/the baseline signal value *100 as described in our previous study [33].

## Figures and Tables

**Figure 1 ijms-25-07233-f001:**
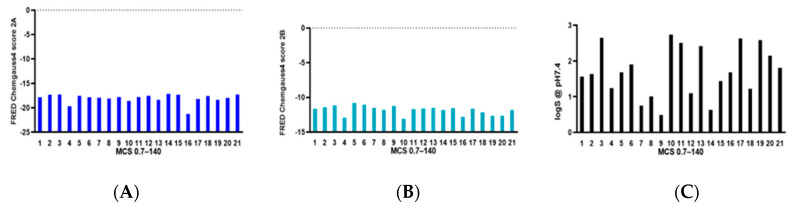
The properties of 140 compounds in 21 categories. The compounds’ affinity to the α2A adrenergic receptor (**A**), α2B adrenergic receptor (**B**), and LogS (**C**) of compounds in 21 categories.

**Figure 2 ijms-25-07233-f002:**
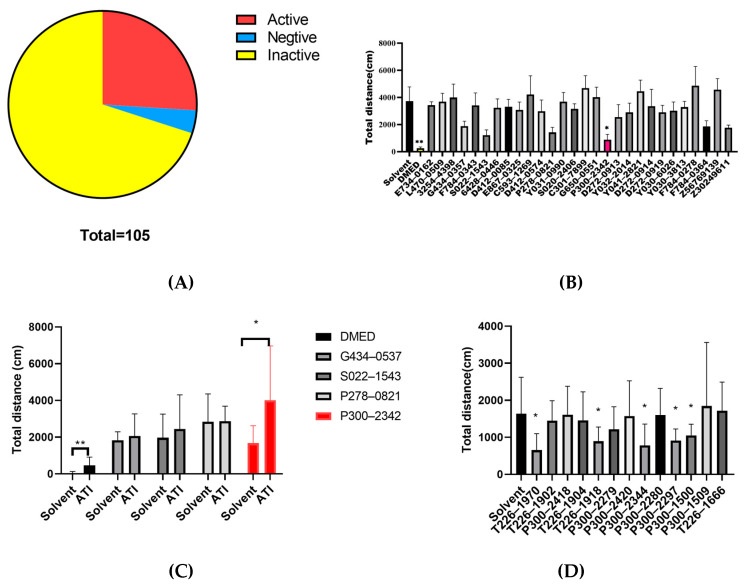
The effects of compounds on the LORR and locomotor activity in mice. (**A**) A total of 105 compounds (100 nmol/mouse, i.c.v.) 15 min prior to DMED administration (432 nmol/kg, i.p.). Twenty-eight compounds increased (active) and four compounds decreased (negative) the percent of LORR induced by DMED (432 nmol/kg, i.p.). Seventy-three compounds displayed no effects (inactive). (**B**) The effect of the active compounds (100 nmol/mouse, i.c.v.) on the locomotor activity of mice. P300–2342 was red. (**C**) The effects of atipamezole (ATI,0.2 mg/kg, i.p.) on the locomotor activity inhibited by four compounds and DMED (10 nmol/mouse, i.c.v.). The red column showed the antagonism of ATI on P300–2342. (**D**) The locomotor activity of mice induced by 13 analogs of P300–2342. *n* = 8, mean ± SEM, * *p*< 0.05, ** *p* < 0.01 compared with the solvent. The analysis was performed using a one-way ANOVA followed by Dunnett’s test (**B**,**D**) and a two-way ANOVA followed by Sidak’s test (**C**).

**Figure 3 ijms-25-07233-f003:**
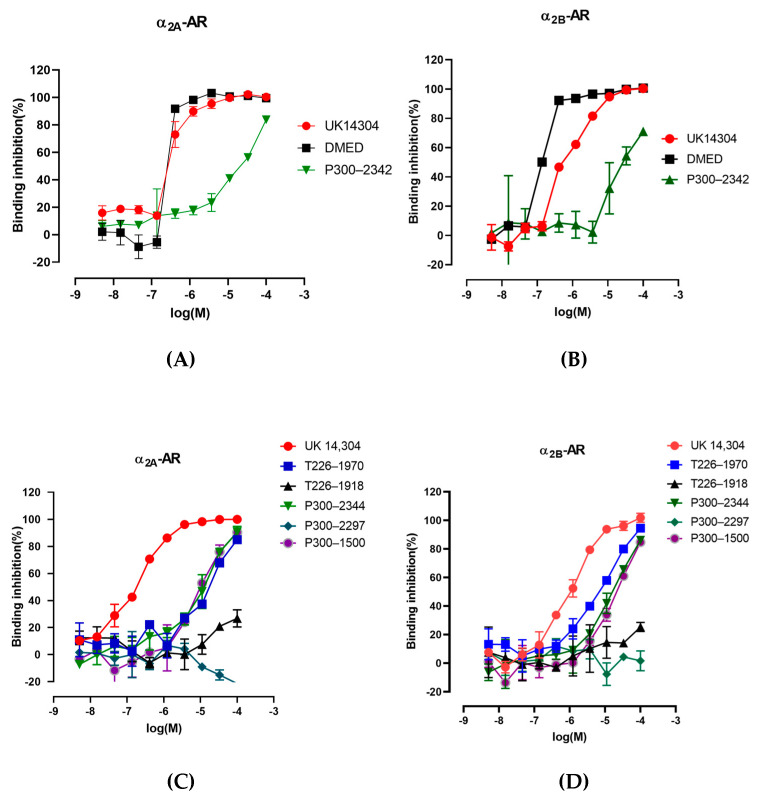
Binding affinities of P300–2342 and its analogs for human α_2A_ and α_2B_-AR. (**A**) Binding affinities of P300–2342 compared with α_2_-AR agonists UK14304 and DMED for human α_2A_-AR. (**B**) Binding affinities of P300–2342 compared with α_2_-AR agonists UK14304 and DMED for human α_2B_-AR. (**C**) Binding affinities of P300–2342 analogs (T226–1970, T226–1918, P300–2344, P300–2297,P300-1500) compared with α_2_-AR agonists UK14304 for human α_2A_-AR. (**D**) Binding affinities of P300–2342 analogs (T226–1970,T226–1918,P300–2344,P300–2297,P300-1500) compared with α_2_-AR agonists UK14304 for human α_2B_-AR.

**Figure 4 ijms-25-07233-f004:**
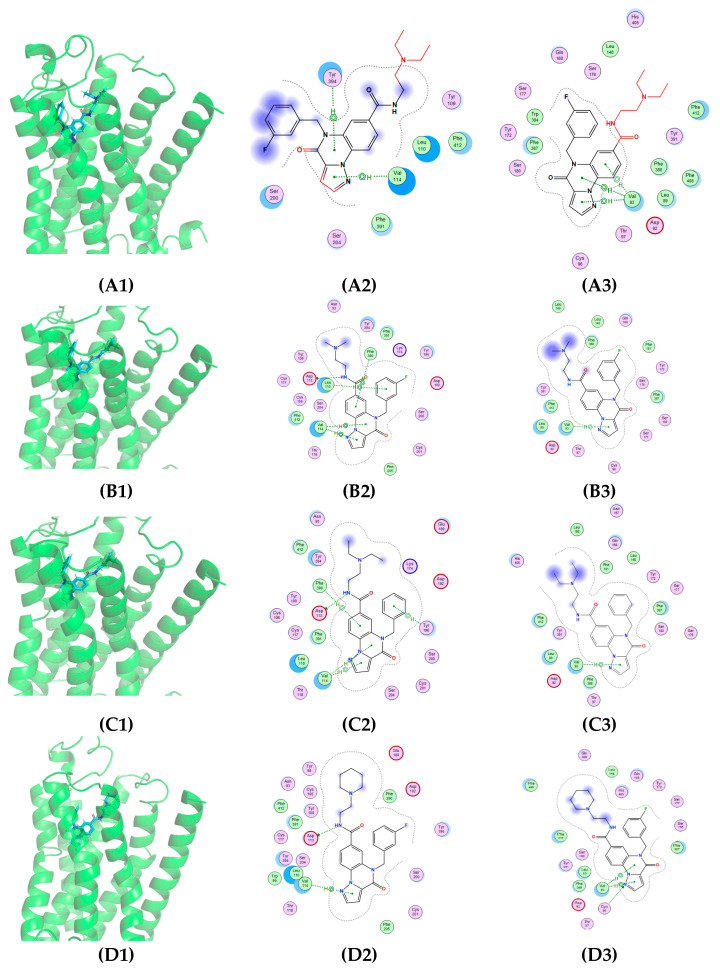
The binding mode of P300–2342 and its analogs to α_2A_ or α_2B_-AR. (**A1**) The binding mode of P300–2342 with α_2A_-AR; (**A2**,**A3**) the diagrammatic form of the active sites of the complex P300–2342 with α_2A_ or α_2B_-AR. (**B1**) The binding mode of T226–1970 with α_2A_-AR; (**B2**,**B3**) ligand interaction diagram of the complex T226–1970 with α_2A_ or α_2B_-AR. (**C1**) The binding mode of P300–1500 with α_2A_-AR; (**C2**,**C3**) the diagrammatic form of the active sites of the complex P300–1500 with α_2A_ or α_2B_-AR. (**D1**) The binding mode of P300–2344 with α_2A_-AR; (**D2**,**D3**) The diagrammatic form of the active sites of the complex P300–2344 with α_2A_ or α_2B_-AR. The hydrogen bonds are indicated by a dotted line. The solvent-accessible surface is indicated by dotted lines. Groups of the compound in the solvent are shown in red. The residual properties are indicated by colors such as purple indicating the polar residues.

**Figure 5 ijms-25-07233-f005:**
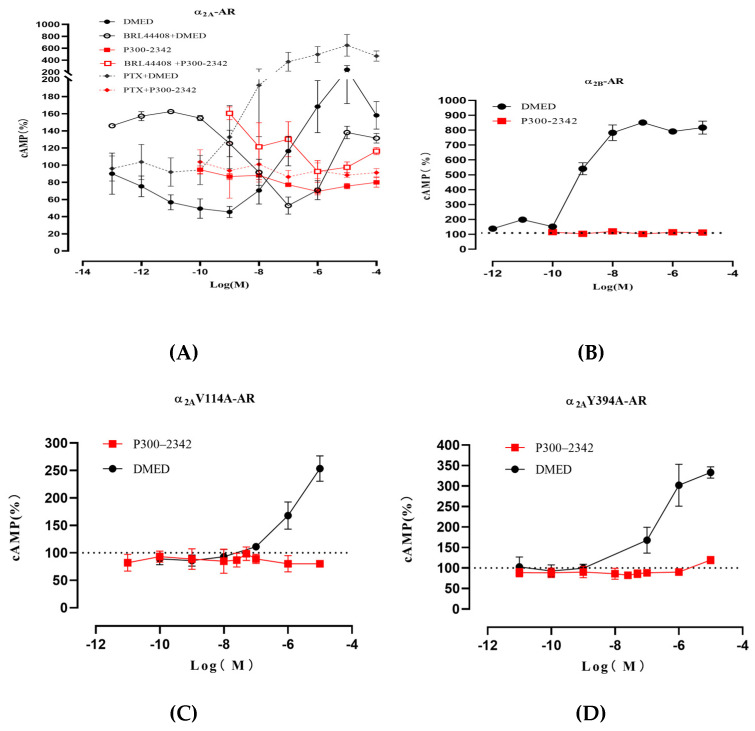
The cAMP production in HEK293 cells transfected with α_2A_-AR, α_2B_-AR, or α_2A_-AR mutants induced by P300–2342. (**A**) HEK293 cells transiently co-transfected with pGloSensor-22F cAMP and pCMV6 Entry-Flag-α_2A_. The production of cAMP was detected by a time-resolved cAMP measurement assay after P300–2342 or DMED addition without or with BRL44408 pretreatment (1 μM). An amount of 500 ng/mL of pertussis toxin (PTX) was pretreated for 24 h to assay the G_αi_ involvement in the inhibition of AC-cAMP accumulation. (**B**) HEK293 cells transiently co-transfected with pGloSensor-22F cAMP and pCMV6 Entry-Flag-α_2B_-AR; the production of cAMP was detected after P300–2342 or DMED addition. HEK293 cells transiently co-transfected with pGloSensor-22F cAMP and pCMV6 Entry-Flag-α_2A_V114A-AR (**C**) or pCMV6 Entry-Flag-α_2A_Y394A-AR (**D**). the production of cAMP was detected by a time-resolved cAMP measurement assay after P300–2342 or DMED addition; data are shown as the mean ± S.E.M value of three independent experiments performed in triplicate.

**Figure 6 ijms-25-07233-f006:**
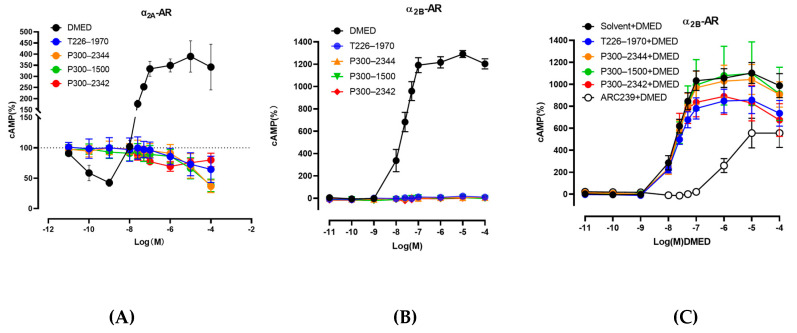
The cAMP production in HEK293 cells transfected α_2A_-AR or α_2B_-AR induced by P300–2342 analogs. (**A**) HEK293 cells transiently co-transfected with pGloSensor-22F cAMP and pCMV6 Entry-Flag-α_2A_-AR; the production of cAMP was detected by a time-resolved cAMP measurement assay after T226–1970, P300–2344, P300–1500, P300–2342, or DMED addition. (**B**) HEK293 cells transiently co-transfected with pGloSensor-22F cAMP and pCMV6 Entry-Flag-α_2B_-AR. The production of cAMP was detected by a time-resolved cAMP measurement assay after T226–1970, P300–2344, P300–1500, P300–2342, or DMED addition. (**C**) The effects of T226–1970, P300–2344, P300–1500, and P300–2342 pretreatment on the cAMP accumulation induced by DMED in pGloSensor-22F cAMP and pCMV6 Entry-Flag-α_2B_-AR co-transfected cells.

## Data Availability

Data are contained within the article and Appendix A.

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
