# Peer review of "The Discovery of Novel α2a Adrenergic Receptor Agonists Only Coupling to Gαi/O Proteins by Virtual Screening"

_ijms, 2024, doi:10.3390/ijms25137233_

Round 1
Reviewer 1 Report
Comments and Suggestions for Authors
1. In page 10: line 327 and 329: what is the resolution of the selected pdbs 6kuy and 6k42?
2. Any co-crystallized ligands present in the selected pdbs 6kuy and 6k42?
3. Provide references for LORR and locomotor activity, Ligand binding assay etc.
4. Any acute toxicity studies are performed before fix the doses?
5. How will you fix the doses for the selected drugs?
6. What are the doses used to find the activities to calculate IC50 values?
7. Which solvent was used to dissolve the compounds for the activity determination?
8. How will yo administer the compounds to the mice?
9. Any standard compound is used to compare the activity?
10.Provide the docking results in a table at least for top active compounds in supplementary data.
11.Provide the ADMET results in a table at least for top active compounds in supplementary data.
12.Provide the expansion of abbreviations for many terms throughout the manuscript.
13.Many references are more than 10-20 years old. Provide some latest references.
Author Response
- In page 10: line 327 and 329: what is the resolution of the selected pdbs 6kuy and 6k42?
Answer: The resolution of the 6kuy is 3.20 Å and the 6k42 is 4.10 Å
- Any co-crystallized ligands present in the selected pdbs 6kuy and 6k42?
Answer: The pdb 6kuy is the crystal structure of the α2A adrenergic receptor in complex with a partial agonist E39 and the 6k42 is the cryo-EM structure of alpha2B AR-Gi in complex with agonist dexmedetomidine.
- Provide references for LORR and locomotor activity, Ligand binding assay etc.
Answer: References for LORR and locomotor activity, cAMP assay were provided in manuscript. Ligand binding assay using α2A-AR or α2B-AR membranes (ES-030-M400UA/ NET722250UC, PerkinElmer), radioactivity of each sample in UNIFILTER-96 GF/B plates (PerkinElmer) was counted in a Microbeta2 (PerkinElmer). The ligand binding assay is different traditional ligand binding in our previous study [1]. We had no reference in the ligand binding assay.
[1] Yu, G., Li, S.H., Cui, M.X., Yan, L.D., Yong, Z., Zhou, P.L., Su, R.B., Gong, Z.H., 2014. Multiple mechanisms underlying the long duration of action of thienorphine, a novel partial opioid agonist for the treatment of addiction. CNS Neurosci. Ther. 20,282–288
- Any acute toxicity studies are performed before fix the doses?
Answer: The chemicals were bought from Chemdiv. There was only 100mg P300-2342 in Chemdiv. We got all of the P300-2342 in Chemdiv. In the CCK8 test, P300-2342 (1,10,100μM) had no toxicity in the cells for 48h. We will synthetize P300-2342 and its analogs for acute toxicity in mice.
- How will you fix the doses for the selected drugs?
Answer: The potent and high selective agonist of α2-adrenergic receptors dexmedetomidine(DMED, 10nmol/5μl, i.c.v.) or (432 nmol/kg, i.p.) induced the 50% LORR in mice. To test the sedation effects of the highly selective α2A-AR chemicals by virtual screening, the chemicals (100nmol/5μl, i.c.v.) was 10 times of dexmedetomidine (10nmol/5μl) . In order to minimize the number and harm of animals, compounds (100 nmol/mouse, i.c.v.) pretreatment 15 min prior to the DMED (432 nmol/kg, i.p.) administrated. The compounds increasing the percent of LORR induced by DMED (432 nmol/kg, i.p.) were selected for further study.
- What are the doses used to find the activities to calculate IC50 values?
Answer: The doses were 0.0136, 0.1358,1.358,13.5802,136,407,1222,3667,11000,33000nM in the receptor binding affinity to calculate IC50 values. The horizontal axis in figure 2 was the logarithmic value of the doses.
- Which solvent was used to dissolve the compounds for the activity determination?
Answer: The stock solution of the compounds was 200mM dissolved in DMSO. In figure.2A, the working solution of compounds 20mM was diluted from stock solution using water and the solvent control was 10%DMSO. In figure.2C/2D, the working solution of compounds 2mM was also diluted from stock solution using water and the solvent control was 1%DMSO.
- How will yo administer the compounds to the mice?
Answer: The compounds were administrated in the mice through intracerebroventricular (i.c.v.) injection with volume 5μl.
- Any standard compound is used to compare the activity?
Answer: The potent and high selective agonist of α2-adrenergic receptors DMED was used as standard in mice and cell tests.
10.Provide the docking results in a table at least for top active compounds in supplementary data.
Answer: We provided the docking scores of the active compounds in the supplementary table1.
11.Provide the ADMET results in a table at least for top active compounds in supplementary data.
Answer: We provided the ADMET properties such as weight、logS、logP、HIA、2D6、PPB of the active compounds in the supplementary table1.
12.Provide the expansion of abbreviations for many terms throughout the manuscript.
Answer: We provided the expansion of abbreviations in the resubmitted manuscript.
13.Many references are more than 10-20 years old. Provide some latest references.
Answer: We added several recent references. Most of the research on the structure and function of α 2-AR was conducted 10 years ago. More and more studies have begun after the α 2-AR crystal structure reported in recent years,
Reviewer 2 Report
Comments and Suggestions for Authors
The authors present a very interesting analysis regarding the inhibition of a2 adrenergic receptors. Their data show the value of virtual screening and how it can be implemented to optimize the results. The paper could be improved significantly by addressing the following issues.
In the results section, lines 74-76: the authors report that they have docked 50 million compounds to A2A receptor and used the 100,000 top results of this process to dock them to A2B receptor. What was the rationale behind this choice? Based on the comments by the authors in the paper the aim was to identify selective inhibitors of A2A receptor.
Additionally in Figure 1 the authors present the properties of the different compounds in 21 categories. In the respective text, lines 102-107, the authors highlight the differences between different clusters. But based on figure 1 the text is confusing. In all the graphs in the figures the y axis represents counts (I hypothesize of the compounds) and not the differences in clusters as described in the text. This is a confusing section and needs to be presented in a much more clear manner.
In line 154 the authors state that they have employed a flexible docking protocol to investigate the mechanism of action of the compound. In the next line (155-156) the authors report the rigid docking score. Why is that? Flexible docking is providing the respective scores, and they should be presented in the paper.
While in results section the authors state that they have docked only the 100.000 compounds to A2B receptor (lines 74-76), in discussion (lines 241-246) the authors report that they have docked all the compounds in both receptors. Please clarify the issue.
In the Methods section 4.2 (line 337) the authors report the use of homology models (line 338). In the previous paragraph there is mention of the use of crystal structures. Did the authors perform homology modeling for the proteins? What was the rationale behind this choice? Moreover, if homology modeling was performed there should be a relevant section describing the Methods employed and a relevant section in results commenting on the outcome.
A minor comment is that conclusions should not be present after the Methods section but before it.
Author Response
Answer to reviewer 2
- In the results section, lines 74-76: the authors report that they have docked 50 million compounds to A2A receptor and used the 100,000 top results of this process to dock them to A2B receptor. What was the rationale behind this choice? Based on the comments by the authors in the paper the aim was to identify selective inhibitors of A2A receptor.
Answer: This paper want to obtain highly selective α2A-AR agonists as potential sedatives. Firstly, the top 100,000 compounds with high affinity to α2A-AR were then docked to α2B-AR. The compounds with high affinity to α2B-AR were not selected. Compounds with high affinity to α2A-AR (−22 to −16 kcal/mol) and low affinity to α2B-AR (−14 to −8 kcal/mol) were selected( lines 83 ). We studied 105 compounds with good druggability of the virtual screening chemicals in the lost of righting reflex of C57/B6 mice. After intraventricular administration, 28 compounds increased and 4 compounds decreased the percent of LORR induced by DMED (432 nmol/kg, i.p.). 73 compounds displayed no effect on the LORR induced by DMED (432 nmol/kg, i.p.) [Fig. 2A]. The 28 compounds were further studied in locomotor activity. Among these 28 compounds, 4 compounds significantly inhibited the locomotor activity of mice (P< 0.05) [Fig. 2B]. The aim was to identify selective agonists of α2A-AR.
- Additionally in Figure 1 the authors present the properties of the different compounds in 21 categories. In the respective text, lines 102-107, the authors highlight the differences between different clusters. But based on figure 1 the text is confusing. In all the graphs in the figures the y axis represents counts (I hypothesize of the compounds) and not the differences in clusters as described in the text. This is a confusing section and needs to be presented in a much more clear manner.
Answer: Thanks. Figure 1 were revised according to the suggestion of reviewer. These picture might help to present the properties of the different compounds.
- In line 154 the authors state that they have employed a flexible docking protocol to investigate the mechanism of action of the compound. In the next line (155-156) the authors report the rigid docking score. Why is that? Flexible docking is providing the respective scores, and they should be presented in the paper.
Answer: We are sorry for our carelessness. Flexible docking protocol was a mistake in line 154. we did the virtual screening by using OpenEye software. In our resubmitted manuscript, this mistake is corrected.
- While in results section the authors state that they have docked only the 100.000 compounds to A2B receptor (lines 74-76), in discussion (lines 241-246) the authors report that they have docked all the compounds in both receptors. Please clarify the issue.
Answer: Thanks. We added the top 100,000 were then docked to α2B-AR crystall structures in our resubmitted manuscript.
- In the Methods section 4.2 (line 337) the authors report the use of homology models (line 338). In the previous paragraph there is mention of the use of crystal structures. Did the authors perform homology modeling for the proteins? What was the rationale behind this choice? Moreover, if homology modeling was performed there should be a relevant section describing the Methods employed and a relevant section in results commenting on the outcome.
Answer: We are sorry for our carelessness. At the beginning of the study, there was no crystal structure of human alpha2 (6KUY). After the homology modeling for virtual screening, the crystal structures of α2A-AR and α2B-AR were released. We then adjusted the virtual screening through crystal structure.
- A minor comment is that conclusions should not be present after the Methods section but before it.
Answer: Thanks. We adjusted the conclusions before the methods section in the resubmitted manuscript.
Reviewer 3 Report
Comments and Suggestions for Authors
To Authors
a2A adrenergic receptors couple to Gi/Go and Gs. In this paper, we narrowed down the list of agonists that selectively activate Gi/Go by virtual screening, and then evaluated them by loss of righting reflex (LORR), locomotor activity, and cAMP assay to find compound P300-2342. Although the method shows one direction, there are several problems.
1. α2A adrenergic receptor couples to Gi/Go and Gs. The coupling should be measured directly by BRET or FRET. These methods are more common. Using these methods, it is easy to demonstrate whether compound P300-2342 selectively activates Gi/Go.
2. Weak inhibition of cAMP production by compound P300-2342 and weak effect of pertussis toxin in Figure 5A.
3. In Figure 5A, the 100-200 range should be enlarged.
4. In Figure 2D, the effects of 13 compounds on locomotor activity were measured at N=8. If 8 animals were used for each compound, that means 104 animals were used. It is necessary to be stated that the same animal individuals are not used in this experiment.
5. In Figure 2 legend, "Dunnett's test (A, D)" may be mistaken for "Dunnett's test (B, D)".
6. The letters in the figure are too small and difficult to distinguish. The font size should be enlarged.
Author Response
- α2A adrenergic receptor couples to Gi/Go and Gs. The coupling should be measured directly by BRET or FRET. These methods are more common. Using these methods, it is easy to demonstrate whether compound P300-2342 selectively activates Gi/Go.
Answer: The measure of receptor coupling to Gi/Go or Gs by BRET or FRET is a good method. In our lab, we could not establish the BRET or FRET-based techniques that allow detection of ligand-dependent interactions between specific receptors and specific G proteins. Adenylyl cyclases (AC) converts ATP to cAMP and is activated or inactivated by interaction with Gi/Go or Gs protein respectively [1]. α2A-AR has a dual pharmacological effect in that it simultaneously couples to Gi and Gs to inhibit or stimulate AC activity [2,3]. Similar to these reports, α2A-AR simultaneously coupled to both Gi and Gs proteins when stimulated by agonist DMED in the present study (Fig.5A). The second messenger cAMP plays important role in sedation after α2A-AR activation. Pre-treatment with phosphodiesterase 4 inhibitor rolipram or dibutyryl-cAMP (dbcAMP) inhibited the mice LORR induced by DMED in our previous study[4]. GloSensor cAMP biosensor (Promega) manufacturer's protocols is very easy to detect the intracellular cAMP after HEK293 cells co-transfected with the pGloSensor-22F cAMP plasmid and pCMV6 Entry-Flag-α2A-AR. This method always used in the detection of GPCR activation like opioid receptors, 5-HT receptors et al after the compounds addition. In addition to the second messenger cAMP accumulation in cells, the effects of compounds on the loss of righting reflex (LORR), locomotor activity in mice were studied to comfirm the α2-AR activation.
[1] Hanoune, J., Defer, N., 2001. Regulation and role of adenylyl cyclase isoforms. Annu. Rev. Pharmacol. Toxicol. 41, 145–174.
[2] Eason, M.G.; Liggett, S.B. Identification of a Gs coupling domain in the amino terminus of the third intracellular loop of the alpha 2A-adrenergic receptor. Evidence for distinct structural determinants that confer Gs versus Gi coupling. J Biol Chem. 1995, 270, 24753-24760
[3] Eason, M.G.; Kurose, H.; Holt, B.D.; Raymond, J.R.; Liggett, S.B. Simultaneous coupling of alpha 2-adrenergic receptors to two G-proteins with opposing effects. Subtype-selective coupling of alpha 2C10, alpha 2C4, and alpha 2C2 adrenergic receptors to Gi and Gs. J Biol Chem. 1992, 267, 15795–15801.
[4]Liu, M.; Yang, Y.; Tan, B.; Li, Y.L.; Zhou,P.L.; Su, R.B. G αi and G βγ subunits have opposing effects on dexmedetomidine-induced sedation. Eur J Pharmacol. 2018, 831, 28-37.
- Weak inhibition of cAMP production by compound P300-2342 and weak effect of pertussis toxin in Figure 5A.
Answer: The maximum inhibiton of cAMP production was about 30% after P300-2342 (1μM) treatment. The maximum inhibition of cAMP production was about 55% after the potent α2-AR agonist of DMED (10nM) addition. The maximum inhibiton of cAMP production induced by P300-2342 was about 50% that of DMED. These results were coincide with the locomotor activity in mice (fig2B). The inhibition of locomotor activity induced by P300-2342(100nmol) was 20% lower that of DMED (10nmol). These results indicated under P300-2342 treatment, the activation of α2-AR coupling to Gi protein was weaker compared with that of DMED. In contrast to DMED, P300-2342 could not activate α2-AR coupling to Gs protein. Pretreatment with Gαi/o inhibitor PTX, The cAMP production was 90-100% of that in control after P300-2342(10-10-10-4M) or DMED (10-13-10-10M) addition. The cAMP production induced by P300-2342 was similar to that of DMED under the pretreatment of Gαi/o inhibitor PTX.
- In Figure 5A, the 100-200 range should be enlarged.
Answer: We enlarged the range in the resubmitted manuscript.
- In Figure 2D, the effects of 13 compounds on locomotor activity were measured at N=8. If 8 animals were used for each compound, that means 104 animals were used. It is necessary to be stated that the same animal individuals are not used in this experiment.
Answer: In pharmacology tests, each animal is commonly given one dose of chemicals or one treatment and will not be reused. The locomotor activity induced by each compound was detected in 8 mice. It is generally not need to be stated. Our previous published articles have not stated the animal individuals[1] .
[1]Liu, M.; Yang, Y.; Tan, B.; Li, Y.L.; Zhou,P.L.; Su, R.B. G αi and G βγ subunits have opposing effects on dexmedetomidine-induced sedation. Eur J Pharmacol. 2018,831,28-37.
- In Figure 2 legend, "Dunnett's test (A, D)" may be mistaken for "Dunnett's test (B, D)".
Answer: Thanks. We adjusted the mistaken in figure 2 in the resubmitted manuscript.
- The letters in the figure are too small and difficult to distinguish. The font size should be enlarged.
Answer: We enlarged the font size in the figure.
Round 2
Reviewer 2 Report
Comments and Suggestions for Authors
The authors seem to have addressed all the issues that have been highlighted in the initial report in a carefull and concise manner. Therefore I do not have any other comments regarding the paper.
Reviewer 3 Report
Comments and Suggestions for Authors
Due to the measuring equipment, authors are unable to measure the coupling between the receptor and the G protein downstream of the G protein, but otherwise the resubmitted paper corresponds adequately. Therefore, there is no further comments.
